# Detecting recurrent passenger mutations in melanoma by targeted UV damage sequencing

Kathiresan Selvam ®[1], Smitha Sivapragasam ®[1], Gregory M. K. Poon ®[2] & John J. Wyrick ®[1,3] ✉

Sequencing of melanomas has identified hundreds of recurrent mutations in both coding and non-coding DNA. These include a number of well-characterized oncogenic driver mutations, such as coding mutations in the *BRAF* and *NRAS* oncogenes, and non-coding mutations in the promoter of telomerase reverse transcriptase (*TERT*). However, the molecular etiology and significance of most of these mutations is unknown. Here, we use a new method known as CPD-capture-seq to map UV-induced cyclobutane pyrimidine dimers (CPDs) with high sequencing depth and single nucleotide resolution at sites of recurrent mutations in melanoma. Our data reveal that many previously identified drivers and other recurrent mutations in melanoma occur at CPD hotspots in UV-irradiated melanocytes, often associated with an overlapping binding site of an E26 transformation-specific (ETS) transcription factor. In contrast, recurrent mutations in the promoters of a number of known or suspected cancer genes are not associated with elevated CPD levels. Our data indicate that a subset of recurrent protein-coding mutations are also likely caused by ETS-induced CPD hotspots. This analysis indicates that ETS proteins profoundly shape the mutation landscape of melanoma and reveals a method for distinguishing potential driver mutations from passenger mutations whose recurrence is due to elevated UV damage.

A distinguishing characteristic of many oncogenic mutations is that they reoccur at the same genomic position in independent tumors. For example, somatic mutations in the V600 codon of the *BRAF* oncogene (i.e., *BRAF* V600E or V600K) occur in as many as 50% of melanomas, consistent with data indicating that these oncogenic driver mutations promote cell proliferation and carcinogenesis[1–7]. Recurrent somatic mutations have been detected not only in other oncogenes (e.g., *NRAS*, etc.), but also in non-coding DNA[1,2,4,8,9]. Recurrent non-coding mutations have been identified at two primary locations in the promoter of the human telomerase reverse transcriptase (*TERT*) gene in melanomas and other cancers[10–12], both of which up-regulate *TERT* expression

and telomerase activity[13–15] by creating a binding site for E26 transformation-specific (ETS) family transcription factors (TF). For these reasons, mutational recurrence is often viewed as strong evidence for driver function, both in the literature and in driver prediction software[8,16,17].

Analysis of cohort of 183 sequenced melanoma genomes has revealed more than a 100 other recurrent somatic mutations in both coding and non-coding DNA[4]. While a few of these have been previously suggested to function as driver mutations specific to melanoma or other skin cancers[4,18,19], the vast majority are uncharacterized. Mutations in melanoma are principally caused by UV-induced DNA

[1]School of Molecular Biosciences, Washington State University, Pullman, WA 99164, USA. [2]Department of Chemistry and Center for Diagnostics and Therapeutics, Georgia State University, Atlanta, GA 30303, USA. [3]Center for Reproductive Biology, Washington State University, Pullman, WA 99164, USA. ✉ e-mail: jwyrick@wsu.edu

damage[4,20–24], primarily cyclobutane pyrimidine dimers (CPDs) that form between neighboring pyrimidine bases (i.e., dipyrimidines). Genome-wide maps of CPDs in UV-irradiated cells[25–29] has revealed that CPD lesions are greatly elevated at binding sites of ETS transcription factors[30]. A number of recurrent non-coding mutations in melanomas are also located in predicted ETS binding sites[9,10,25,26,31,32], leading to the hypothesis that many of these may be passenger mutation hotspots, whose recurrence is due to elevated levels of local UV damage, not carcinogenic selection[9,26,31]. However, testing this hypothesis at individual ETS binding sites has been challenging, because the vast number of potential CPD lesions sites in the human genome (>1.5 billion dipyrimidine sequences) results in very low sequencing depth at any particular site. Hence, it is currently impossible to distinguish whether recurrent mutations at ETS binding sites or other genomic features in melanoma represent bona fide driver mutations, as is the case for promoter mutations upstream of *TERT*[14] and potentially other genes

(e.g., *SDHD*[33]), or are simply a consequence of a high local mutation rate due to an ETS-induced UV damage hotspot.

## Results

### Targeted UV damage sequencing maps CPD hotspots in primary melanocytes

We developed the CPD-capture-seq method (Fig. 1a) to analyze UV damage with high sequencing depth and single nucleotide resolution at individual ETS-binding sites and other recurrently mutated regions in the human genome. The CPD-capture-seq method differs from our published CPD-seq method[34] in that it includes a target capture step prior to library sequencing to enrich for genomic regions of interest (Fig. 1a). We captured genomic regions containing active ETS binding sites (~3000 sites; see Methods), or regions containing recurrent mutations in melanoma[4], located in transcription factor binding sites, promoter regions, untranslated regions (UTR) or coding regions of

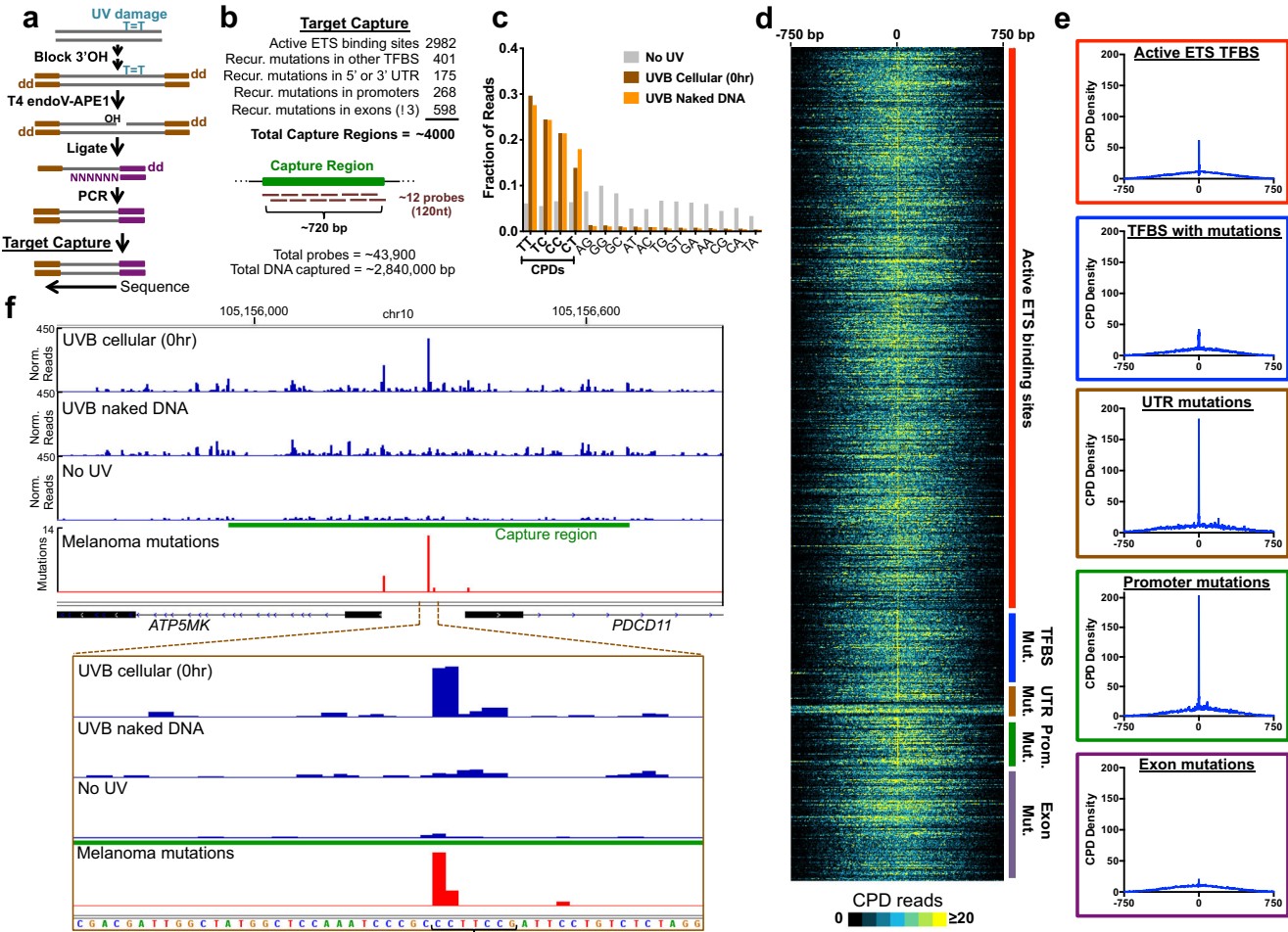

**Fig. 1 | CPD-capture-seq maps UV damage at high sequencing depth and single nucleotide resolution at targeted genomic regions. a** Schematic of CPD-capture-seq protocol. UV-damaged DNA is ligated to first adapter (brown) which contains a dideoxy 3′ end on one side of the adapter. CPD lesions are cleaved with the repair enzymes T4 endonuclease V (T4 endo V) and APE1, and the resulting 3′ hydroxyl (3′OH) is ligated to the second adapter (purple). Genetic regions of interest are selected from the resulting sequence library by a capture step prior to sequencing. **b** List of genomic regions selected for capture. Recurrent somatic mutations in melanoma in different genomic classes were taken from Hayward et al.[4]. Each capture region consisted of typically at least 720 bp of genomic sequence that was tiled by ~11–12 (or more) 120 nt probes. **c** CPD-capture-seq reads in UV-irradiated samples are enriched at CPD-forming dipyrimidine sequences. Fraction of CPD-capture-seq reads associated with a putative lesion at the indicated dinucleotide in UVB-irradiated melanocytes, UVB-irradiated melanocyte genomic DNA, and

un-irradiated melanocyte DNA. **d** Cluster plot showing distribution of CPD lesions in UVB-irradiated melanocytes detected by CPD-capture-seq in each targeted genomic region, sorted by region class. Color indicates number of CPD lesions detected. Image generated using Treeview[75]. **e** Graphs showing CPD density in UVB-irradiated melanocytes detected by CPD-capture-seq in each targeted region class are shown in **d**. **f** CPD hotspots in the promoter of *PDCD11* and *ATP5MK* in UVB-irradiated melanocytes (UVB cellular [0 hr]) coincide with locations of somatic mutation hotspots in sequenced melanoma genomes. Normalized CPD-capture-seq reads (associated with lesion-forming dipyrimidine sites) are shown for UVB-irradiated melanocytes, UVB-irradiated naked genomic DNA, and an un-irradiated (No UV) genomic DNA sample. Data normalized using total CPD-capture-seq read counts for each library. Image generated using the Integrative Genomics Viewer (IGV)[74]. Source data for graphs in **c** and **e** are provided as a Source Data file.

genes (Fig. 1b). Each capture region consisted of ~720 base pairs, which were typically tiled by 11–12 (or more) overlapping 120 nucleotide probes. Altogether, a total of ~4000 genomic regions were captured, representing nearly 3 Mbp of genomic sequence (Fig. 1b).

CPD-capture-seq was used to map CPD lesions in primary human melanocytes immediately after exposure to UVB irradiation (Cellular 0 hr). As controls, CPD-capture-seq was also used to map CPD lesions in genomic DNA from un-irradiated cells (No UV) and from isolated melanocyte genomic DNA irradiated in vitro (Naked DNA). CPD-capture-seq reads from the UVB-irradiated samples were almost entirely associated with CPD-forming dipyrimidine sequences, while no enrichment was observed for the No UV control (Fig. 1c). Visualization of lesions detected by the CPD-capture-seq data from UVB-irradiated melanocytes revealed high lesion density across nearly all the captured genomic loci, with lesion density extending ~300–400 bp in each direction from the center of the capture region (Fig. 1d, e). Notably, the center of the capture regions had a narrow peak of damage in the ETS capture regions, coinciding with the location of the ETS binding site in these regions (Fig. 1d, e). A peak of damage is also visible in the center of the capture regions in transcription factor binding sites associated with recurrent mutations (e.g., ETS family, CTCF, etc.), as well as recurrent mutations in 5′ and 3′ UTRs and promoter regions (Fig. 1d, e). However, a damage peak is largely absent from recurrent mutations in coding exons (Fig. 1d, e). Examination of one of the capture regions associated with a recurrent melanoma mutation in the promoter of *PDCD11*[31] revealed a very strong damage peak in the UVB-irradiated melanocyte sample, exactly coinciding with the location of the recurrent mutation (Fig. 1f). In contrast, no damage induction was observed in the Naked DNA or No UV controls. Both the mutation and damage hotspot were associated with a putative ETS binding site (Fig. 1f). A smaller mutation hotspot was observed near the transcription start site (TSS) of the adjacent *ATP5MK* gene (Fig. 1f and Supplementary Fig. 1a), which also coincided with a CPD peak specifically in UVB-irradiated cells and was associated with an ETS binding motif (Supplementary Fig. 1a). Notably, a compilation of four published CPD-seq libraries of UV-irradiated human skin cells[25] had a much lower density of reads in this genomic region (Supplementary Fig. 1b, c), highlighting the importance of the capture step.

### UV damage in melanocytes is induced at a subset of ETS binding sites that coincide with mutation hotspots in melanoma

We used CPD-capture-seq to examine CPD levels at active ETS binding sites (defined as a ChIP-seq binding site for ETS family members ETS1, ELK4, or GABPA from ENCODE[35] associated with a DNase I hypersensitivity site in primary melanocytes[25,36,37]). Analysis of canonical ETS binding sites revealed damage induction in cellular DNA relative to the naked DNA control at the TC and CC base steps, corresponding to positions −1/0 and 0/+1 in the ETS motif (Fig. 2a). These damage hotspots coincided with the locations of elevated somatic mutation density in 183 sequenced melanoma genomes. A subset of ETS binding sites, including ETS motifs located in the *PDCD11/ATP5MK* promoter, have a dipyrimidine sequence at positions −3/−4 from the ETS motif midpoint. Analysis of CPD-capture-seq data at these binding site variants revealed ~7-fold higher CPD levels at positions −3/−4 in UVB-irradiated melanocytes relative to the naked DNA control, and ~4-fold higher than positions −1/0 in the ETS motif (Fig. 2b). The −3/−4 CPD hotspot coincided with very high rates of somatic mutations in melanoma at these positions (Fig. 2b), which were enriched ~60-fold higher than the expected mutation frequency, based on tri-nucleotide DNA sequence context. Analysis of an independent set of CPD-capture-seq experiments, derived from UVC-irradiated primary melanocytes (Supplementary Fig. 2a) showed a similar pattern of damage induction at ETS binding sites (Supplementary Fig. 3a, b), which closely resembled results from previous CPD-seq libraries[25,38] (Supplementary Fig. 4a, b). CPD density at the −3/−4 position of ETS binding site

variants was not as highly induced following UVC irradiation (~4-fold) as was observed with UVB (Supplementary Fig. 3b), potentially due to UVC-induced photoreversion of CPDs[39].

We used the CPD-capture-seq data to visualize CPD induction at individual ETS binding sites by analyzing the difference in CPD-capture-seq reads between UVB-irradiated melanocytes (cellular) and the scaled naked DNA control (Fig. 2c, left panel and Supplementary Fig. 5). CPD induction was primarily observed at position −3/−4 (for variant binding sites; see Fig. 2c) and positions −1/0 and 0/+1 in the ETS motif. However, even after removing binding sites with weak capture efficiency (see Methods), only two-thirds (or fewer) of the ETS binding sites showed CPD induction relative to the naked DNA control (Fig. 2c and Supplementary Fig. 5). Closer inspection revealed that a similar set of variant ETS binding sites showed CPD induction in the UVB- and UVC-irradiated melanocytes (Supplementary Fig. 3c).

To estimate the significance of CPD induction, we calculated the average difference in CPD counts in UVB-irradiated melanocytes relative to the scaled naked DNA control for regions flanking each variant ETS binding site (e.g., 6 to 180 bp away). The average CPD induction in flanking DNA was −1.5 and standard deviation was ~27; similar values were obtained for DNA flanking derived from all CPD-capture-seq regions. Using these values, we calculated the Z-score of CPD induction at variant ETS binding sites. This analysis indicates that many ETS binding sites had CPD induction Z-scores >3 (Supplementary Fig. 6), reflecting CPD induction more than three standard deviations higher than the average. This is likely an underestimate of the Z-score, since CPD induction due to other, non-ETS transcription factors and dipyrimidine-specific differences in CPD induction (e.g., TT versus CC) likely inflate the variance in CPD induction in flanking DNA. In summary, this analysis confirms that ETS binding sites show significant induction of CPDs in UV-irradiated melanocytes.

Somatic mutations in melanoma appeared to correlate with elevated CPD induction at a subset of ETS binding sites in primary melanocytes (Fig. 2c). To more rigorously test this hypothesis, we used a Poisson regression model to predict melanoma mutation counts at positions −3/−4 in variant ETS binding sites using the CPD-capture-seq data from UVB-irradiated melanocytes and/or the UVB-irradiated naked DNA control. The null model, which only used CPD-capture-seq reads from the UVB-irradiated naked DNA control, was a very poor predictor of mutation counts (pseudo $R^2 < 0.001$) and the naked DNA CPD-capture-seq reads did not significantly correlate with mutation ($P > 0.05$). In contrast, the alternative model, which used CPD-capture-seq reads from both UVB-irradiated melanocytes and the naked DNA control as independent variables, was a significantly better predictor than the null model ($P < 0.0001$ based on likelihood ratio test; pseudo $R^2 = 0.14$). CPD counts from the UVB-irradiated melanocytes showed a significant positive correlation with mutation count (i.e., positive coefficient in regression equation; $P < 0.0001$), while CPD counts in the naked DNA control showed a significant negative correlation (i.e., negative coefficient; $P < 0.0001$). This regression equation indicates that the scaled difference in CPD counts between the cellular and naked DNA control (i.e., CPD induction) at ETS binding sites significantly correlates with mutation count in melanoma.

The lack of damage induction at a subset of ETS sites likely reflects the absence of ETS binding in this particular cell type (primary melanocytes). To test this possibility, we analyzed the local density of DNase-seq reads derived from primary melanocytes[36] at each binding site (see Methods). While all of the ETS binding sites analyzed were associated with a DNase I hypersensitivity site, we reasoned that some of the binding sites might have relative lower DNase-seq reads due to lower site accessibility and/or activity. This analysis indicated that the average local density of DNase-seq reads significantly correlated with the level of CPD induction at variant ETS binding sites (Fig. 2c; Spearman's $\rho = 0.41$ (95% Confidence Interval (CI): 0.3140–0.5007, $P < 0.0001$). These results indicate that CPD induction was higher at

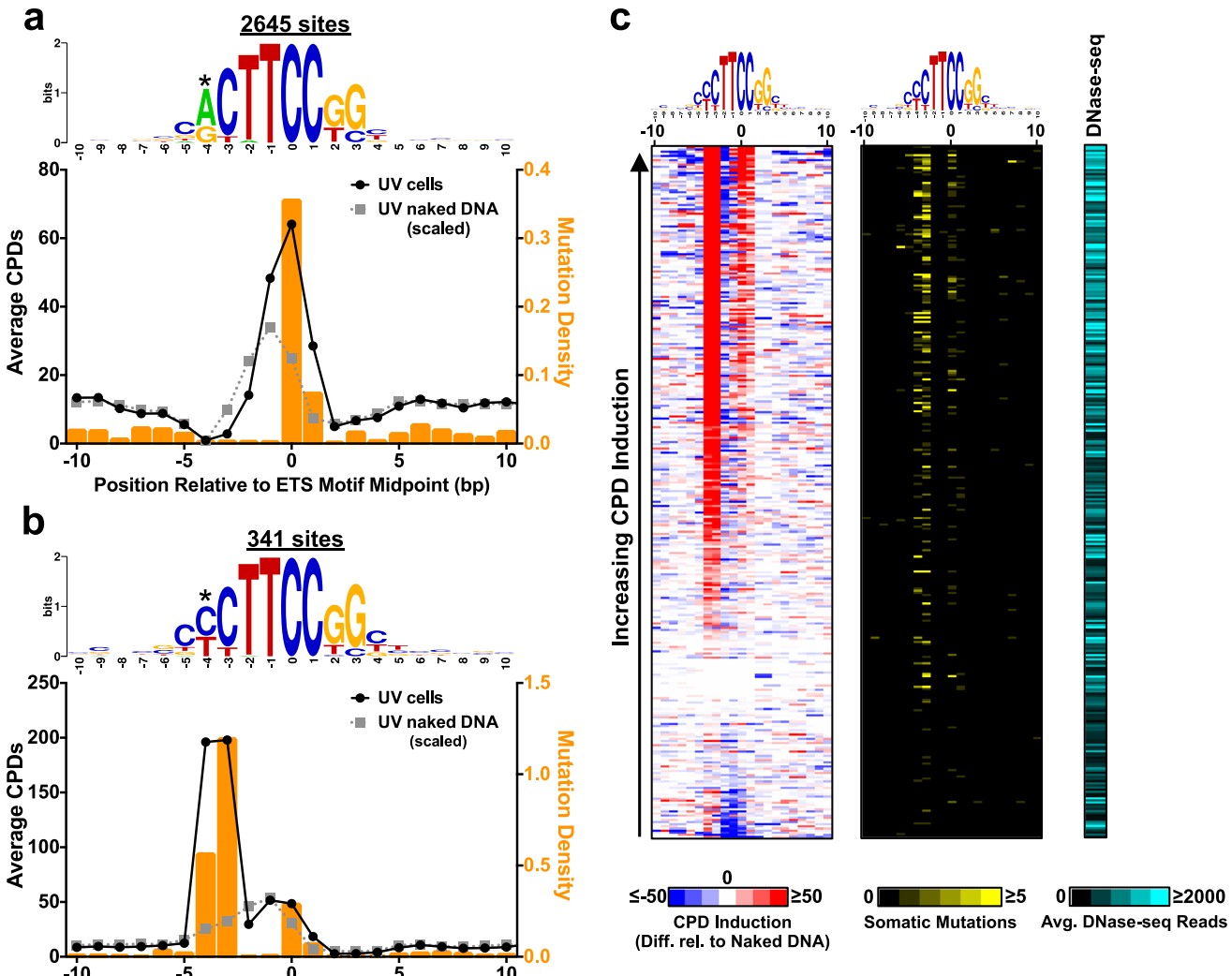

**Fig. 2 | UV-induced CPD lesions are induced at a subset of ETS binding sites, which are associated with somatic mutation hotspots in melanoma. a** Plot of average CPDs per ETS binding site at positions spanning a 20 bp window centered on the midpoint of active ETS binding sites (i.e., ETS-binding sites identified by ENCODE[35] associated with a melanocyte DNase I hypersensitivity region[36]). CPD-capture-seq data from UVB-irradiated melanocytes (UV cells; black circles) and UVB-irradiated naked DNA control (gray squares) is depicted. The CPD-capture-seq data from the naked DNA controls was scaled so that the total number of dipyrimidine-associated reads were equivalent. Somatic mutation density from 183 sequenced melanoma genomes (ICGC) is included for comparison (orange bars). Only canonical ETS binding sites that lack a dipyrimidine at positions -3/−4 relative to the ETS motif midpoint are shown. Sequence logo of ETS binding sites was generated using Weblogo[78]. **b** Same as **a**, except only variant ETS binding sites with a dipyrimidine at positions −3/−4 relative to the ETS motif midpoint are shown. **c** Cluster plot of CPD induction at each individual variant ETS binding site. Color indicates the level of CPD induction, defined as the difference in CPD levels at each position in UVB-irradiated melanocytes relative to the scaled naked DNA control (see color bar). For comparison, the number of somatic mutations at each position in 183 sequenced melanoma genomes (middle panel) and the density of DNase-seq reads in melanocytes within 50 bp of the binding site (right panel) are depicted. In all panels, binding sites are ordered by increasing CPD induction in the ETS motif in UVB-irradiated melanocytes. ETS binding sites that had low capture efficiency, defined as fewer than one lesion site per base pair in DNA flanking the ETS binding site, were excluded from the cluster plot. Source data for graphs in **a** and **b** are provided as a Source Data file.

binding sites associated with elevated DNase-seq reads, presumably because these sites are more likely to be accessible and bound by an ETS transcription factor.

We wondered whether variant ETS binding sites that did not show CPD induction in melanocytes might be bound by ETS proteins and show damage induction in other cell types. To investigate this possibility, we used CPD-capture-seq to map CPD lesions in UVB- and UVC-irradiated normal human skin fibroblasts (NHF1 cells), as well as in isolated NHF1 genomic DNA irradiated in vitro (Supplementary Fig. 2b, c). We observed a very similar pattern of damage induction at ETS binding sites in aggregate in NHF1 cells (Supplementary Fig. 4c–f). Analysis of individual sites revealed that many ETS binding sites show similar damage induction in both primary melanocytes

and fibroblasts (Supplementary Fig. 3c, d), but that there are also a number of binding sites that show consistent differences in damage induction between primary melanocytes and fibroblasts (Supplementary Fig. 3c, d and 7). Taken together, these data indicate that UV damage induction occurs at a subset of ETS binding sites in a cell type-specific manner and correlates with somatic mutation density in melanoma.

To test whether CPD induction at TF binding motifs could be identified de novo in the CPD-capture-seq data, we analyzed the number of CPD-capture-seq reads associated with different hexamer sequence contexts (e.g., AGTCAT, underline indicates the location of CPD lesion) in UVB-irradiated melanocytes relative to the UVB-irradiated naked DNA control (Supplementary Fig. 8a). While most

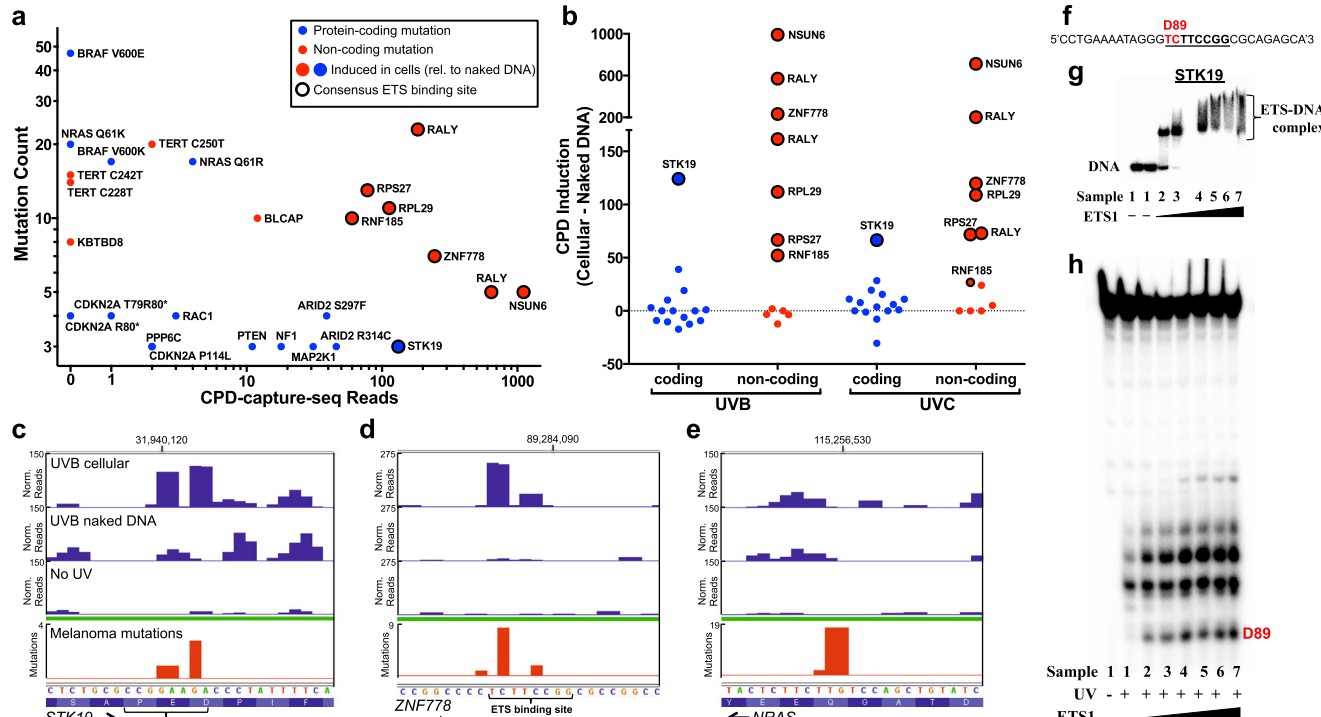

**Fig. 3 | UV damage hotspots at ETS binding sites are associated with, and can potentially explain, a subset of coding and non-coding driver mutations in melanoma. a** Plot of CPD-capture-seq lesions in UVB-irradiated melanocytes and mutation count (from 183 sequenced melanomas) at the indicated mutation sites, which were previously identified as candidate driver mutations in melanoma. Protein-coding driver mutations are in blue; non-coding driver mutations are in red. Larger circles indicate mutation sites associated with significantly elevated CPDs in UVB-irradiated melanocytes relative to scaled UVB-irradiated naked DNA control, defined at least twofold higher and an absolute difference of at least 50. Black outline indicates that the mutation site occurs in an ETS binding site. **b** Same as **a**, except plot of the difference in CPD levels in UVB-irradiated melanocytes relative to scaled UVB-irradiated naked DNA control. **c**–**e** Plot of normalized CPD-capture-seq reads (associated with lesion-forming dipyrimidine sites) for UVB-irradiated melanocytes or naked DNA at **c** *STK19*, **d** *ZNF778* promoter, or **e** *NRAS*. No UV data

are shown as a control. Mutation density derived from 183 sequenced melanoma genomes (ICGC). Images generated using IGV[74]. **f** Oligonucleotide sequence of pyrimidine-containing strand of the *STK19* D89N mutation hotspot (red). ETS-binding motif is indicated in bold underline. *STK19* D89N mutation hotspot is at position −3 relative to the ETS binding motif midpoint. **g** Electrophoretic Mobility Shift Assay (EMSA) shows binding of ETS1 protein to radiolabeled double-stranded *STK19* oligonucleotide (**f**). **h** CPD levels are significantly induced following UV irradiation at the D89 mutation hotspot in the presence of bound ETS1. Denaturing polyacrylamide gel electrophoresis of unbound (sample 1) or ETS1-bound oligo-nucleotides (samples 2–7) with or without UV irradiation, following treatment with T4 endonuclease V, which specifically cleaves at CPD lesions. Only the pyrimidine-containing DNA strand (**f**) is radiolabeled. Representative gels (**g**, **h**) are shown from four independent experiments. Source data for graphs in **a** and **b** are provided as a Source Data file.

hexamers showed roughly similar levels of CPDs in the cellular and naked DNA samples, a few showed striking differences. For example, a number of hexamers that match the ETS binding motif (e.g., CT<u>TC</u>CG, TT<u>CC</u>GG, and TT<u>CC</u>GC) were significantly higher in UV-irradiated cells, consistent with our findings that ETS binding promotes CPD formation at TC and CC dinucleotides in its binding site. There were also elevated cellular CPD levels at sequence contexts that matched the binding consensus (CCAAT/ATTGG) of the Nuclear Factor-Y (NF-Y) TF (e.g., GA<u>TT</u>GG, CA<u>TT</u>GG, TA<u>TT</u>GG, and AA<u>TT</u>GG;). This is consistent with a previous report that NF-Y binding induces CPD formation at a TT sequence in its binding motif[40]. In contrast, a number of sequence contexts had a smaller number of CPD-capture-seq reads in the UV-irradiated cells relative to the naked DNA control (Supplementary Fig. 8a), including hexamers that matched the ETS binding consensus (e.g., CA<u>CT</u>TC, TA<u>CT</u>TC, CG<u>CT</u>TC, and CC<u>CT</u>TC). This is consistent with our findings that ETS binding tends to suppress CPD formation at a CT dinucleotide in its binding site (Fig. 2). We also observed CPD depletion at GACTCA sequences, which match the binding motif of Fos/Jun (i.e., Activator Protein-1, AP-1) TFs. Similar results were obtained when analyzing UVB-irradiated NHF1 cells (Supplementary Fig. 8b). Analysis of active Fos/Jun binding sites that overlapped with CPD-capture-seq regions confirmed CPD depletion in UVB-irradiated cells relative to the naked DNA control (Supplementary Fig. 8c), consistent with our previous report[25]. Taken together, these findings

indicate that CPD-capture-seq data can be used to screen for TF binding sites that modulate CPD formation.

## A subset of putative driver mutations are associated with sites of ETS-induced UV damage

We analyzed CPD-capture-seq data at recurrent driver mutations that had been previously identified in either protein-coding DNA[1,4] or non-coding DNA[4]. We quantified the number of CPDs measured by CPD-capture-seq reads associated with each mutation site. This analysis indicated that most recurrent protein-coding mutations were associated with low CPD levels in UVB-irradiated melanocytes (Fig. 3a). In some cases (e.g., *BRAF* V600E or *NRAS* Q61K) there were no CPDs because the mutation is in a non-dipyrimidine sequence that is unable to form CPD lesions. In contrast, many of the previously identified non-coding driver mutations[4] were associated with very high CPD levels in UVB-irradiated melanocytes, presumably because these recurrent mutations are located in ETS binding motifs (Fig. 3a). Moreover, CPD levels were consistently induced in UVB- or UVC-irradiated melanocytes relative to naked DNA controls, in accordance with ETS binding inducing UV damage formation (Fig. 3b). In contrast, non-coding mutations in the *TERT*, *KBTBD8*, and *BLCAP* promoters, which were not associated with an ETS binding site, showed little to no UV damage or damage induction (Fig. 3a, b). In the case of *TERT*, this may be partly due to poor capture and/or sequencing in this G/C rich genomic region

(Supplementary Fig. 9a, b), but both *KBTBD8* and *BLCAP* also show little to no damage induction, despite efficient capture sequencing in these regions (Supplementary Fig. 9c–f). Non-coding mutations in the *TERT* promoter are well-established driver mutations in melanoma that create functional ETS binding sites[11–15]. Notably, the recurrent promoter mutation in *BLCAP*, a suspected cancer gene[41,42], is also predicted to create an ETS binding site, while *KBTBD8* plays a critical role in melanocyte differentiation[43]. In contrast, candidate non-coding driver mutations associated with high UV damage (and damage induction) primarily occurred in the promoters of housekeeping genes (e.g., ribosomal proteins, etc.) that are unlikely to function in melanomagenesis (Supplementary Table 1). These results suggest that a number of previously identified non-coding driver mutations in melanoma are actually passenger mutations caused by elevated UV damage levels due to ETS binding, which can be detected using CPD-capture-seq.

Notably, there were also relatively high CPD levels associated with a recurrent driver mutation in the coding region of the *STK19* gene (Fig. 3a). This *STK19* D89N mutation was previously identified as an important driver mutation in melanoma[1,44], although the functional consequences of this mutation are controversial[45–48]. Our data indicate that CPD levels at the *STK19* D89 codon are induced in both UVB- and UVC-irradiated melanocytes (relative to the naked DNA controls; Fig. 3b, c), potentially due to binding of an ETS TF to a variant binding sequence that overlaps with the D89 codon (Fig. 3c). Closer inspection revealed significant CPD induction at the −3/−4 and −1/0 positions of the putative ETS binding site in the *STK19* gene (Fig. 3c), which correlated well with mutation hotspots in this gene. This pattern of damage induction was similar to that observed for a recurrent non-coding mutation located in an ETS binding site in the *ZNF778* promoter (Fig. 3d). In contrast, relatively little, if any, CPDs were associated with well-characterized driver mutations in the *NRAS* Q61 codon (Fig. 3e), whose recurrence is clearly due to carcinogenic selection, not damage induction. Our CPD-capture-seq data indicate that CPDs are also induced at *STK19* D89 in UVB-irradiated skin fibroblasts (Supplementary Fig. 10a), suggesting that UV damage is induced at this genomic site in a variety of skin cell types. This may explain why recurrent *STK19* D89N mutations have also been reported in non-melanoma skin cancers[49].

To test whether ETS TFs are able to bind this region of the *STK19* gene and induce UV damage, we purified recombinant ETS1 protein and incubated it with a radiolabeled double-stranded oligonucleotide containing the ETS binding consensus associated with the *STK19* D89 codon (Fig. 3f). Gel shift assays indicated that ETS1 protein binds this DNA sequence in vitro (Fig. 3g). Analysis of CPD lesions following UV irradiation in vitro confirmed that ETS1 binding specifically induces UV damage at the D89 codon (up to 180-fold), as well as at the −1/0 and 0/+1 positions in the ETS binding motif (Fig. 3h). These findings are consistent with our CPD-capture-seq data and indicate that ETS1 (and potentially other ETS family TFs) can bind to this region of *STK19* and promote UV damage.

## Many recurrent non-coding mutations in melanoma are linked to ETS UV damage hotspots

In addition to the candidate non-coding driver mutations mentioned above, more than 100 other recurrent mutations have been identified in promoter regions of sequenced melanoma genomes[4] (defined as ≥5 mutated tumors out of 183 sequence melanomas). Analysis of CPD-capture-seq data revealed that many of these recurrent mutation sites are associated with very high UV damage levels (Fig. 4a; all recurrent promoter mutation sites are shown in Supplementary Fig. 11) that are induced in UVB- and UVC-irradiated melanocytes relative to matched naked DNA controls (Fig. 4b). CPD induction was significantly higher for mutations associated with ETS binding sites (~70% of all recurrent promoter mutations) than for those that were not (Fig. 4a, b). For

example, a previously identified recurrent mutation in the promoter of the *DPH3* gene[9,10,32,50] was associated with very high damage levels in UVB-irradiated melanocytes (Fig. 4a, c), due to damage induction at an ETS binding site in the *DPH3* promoter.

However, not all ETS binding sites show damage induction in UV-irradiated melanocytes. Recurrent mutations in ETS binding sites in the promoters of the known or suspected cancer genes *EGR1*, *ASPSCR1*, and *IQGAP1* genes are not associated with significant damage induction in UVB- or UVC-irradiated melanocytes (Fig. 4a, b). Analysis of a segment of the *EGR1* promoter confirmed that UV damage is not induced at the recurrent mutated ETS binding site (site #1), but is induced at a neighboring ETS site (site #2; see Fig. 4d). Notably, we also observed significant CPD induction at CT dinucleotides in binding sites of the serum response factor (SRF). A roughly similar pattern is apparent in UV-irradiated skin fibroblasts (Supplementary Fig. 10b). The most frequent non-coding mutation in the melanoma cohort occurs in an ETS binding site in the *RPL13A* promoter[4,25], which is also recurrent in other melanoma mutation data sets[10,31]. Surprisingly, we did not observe any damage induction at this site in either primary melanocytes (Fig. 4a, b and Supplementary Fig. 10c, d) or in skin fibroblasts.

In addition to *TERT*, *BLCAP*, and *KBTBD8*, we also observed low levels of damage induction in UV-irradiated melanocytes at a number of other recurrent mutations in the promoters of known or suspected cancer genes (i.e., *TCF3*, *TOP2A*, *NUMB*, *FOSB*, and *OTUB2*; Fig. 4a). In addition to having low damage levels in primary melanocytes, these promoter mutations were not associated with an ETS binding site, suggesting they may be candidate non-coding driver mutations.

## A subset of recurrent protein-coding mutations are linked to ETS-induced UV damage

Since our previous analysis indicated that the *STK19* D89N mutation is associated with (and potentially caused by) elevated UV damage at an overlapping ETS binding site, we performed similar analysis on all recurrent protein-coding mutations in the melanoma cohort (defined as ≥4 mutated tumors out of 183 sequenced melanomas). Many of the recurrent coding mutations were associated with relatively low UV damage, particularly those occurring in known driver genes (Fig. 5a). However, recurrent mutations in three genes (*BCL2L12*, *JMJD8*, and *LTN1*) showed very high UV damage levels, which were significantly induced in both UVB- and UVC-irradiated melanocytes (Fig. 5a, b). Notably, each of these three recurrent mutations were associated with an ETS binding motif, suggesting that UV damage induction may be due to ETS binding. The recurrent mutation in *BCL2L12* results in a synonymous F17F substitution[4,18], which upon closer inspection was confirmed to be associated with UV damage induction at the −1/0 position of an ETS binding motif (Fig. 5c). UV damage was also induced at a neighboring SRF binding sequence, although this damage hotspot was not associated with somatic mutations in melanoma, likely because it occurred at a CT dinucleotide, which is typically not mutagenic. Similarly high damage levels can be observed at the synonymous *JMJD8* L22L and non-synonymous *LTN1* S19F mutation sites (Fig. 5a, b, d), both of which coincided with ETS binding motifs. However, not all ETS motifs in coding regions showed elevated CPD levels in UV-irradiated melanocytes. For example, a recurrent mutation that results in a G34E substitution in one isoform of the *NFKBIE* gene[4,19] was also associated with an ETS binding motif, but this recurrent mutation was not associated with elevated CPD levels (Fig. 5a, b and Supplementary Fig. 12).

Since TF binding normally occurs in promoters and other non-coding DNA, we wondered if there was a common feature that might explain why these particular coding exon sites were targets of ETS TFs. Our analysis indicated that recurrent coding mutations associated with UV damage induction at ETS sites were located close to the transcription start site (TSS) of the gene (median of 400 bp; Fig. 5e). In general, recurrent coding mutations associated with ETS sites were

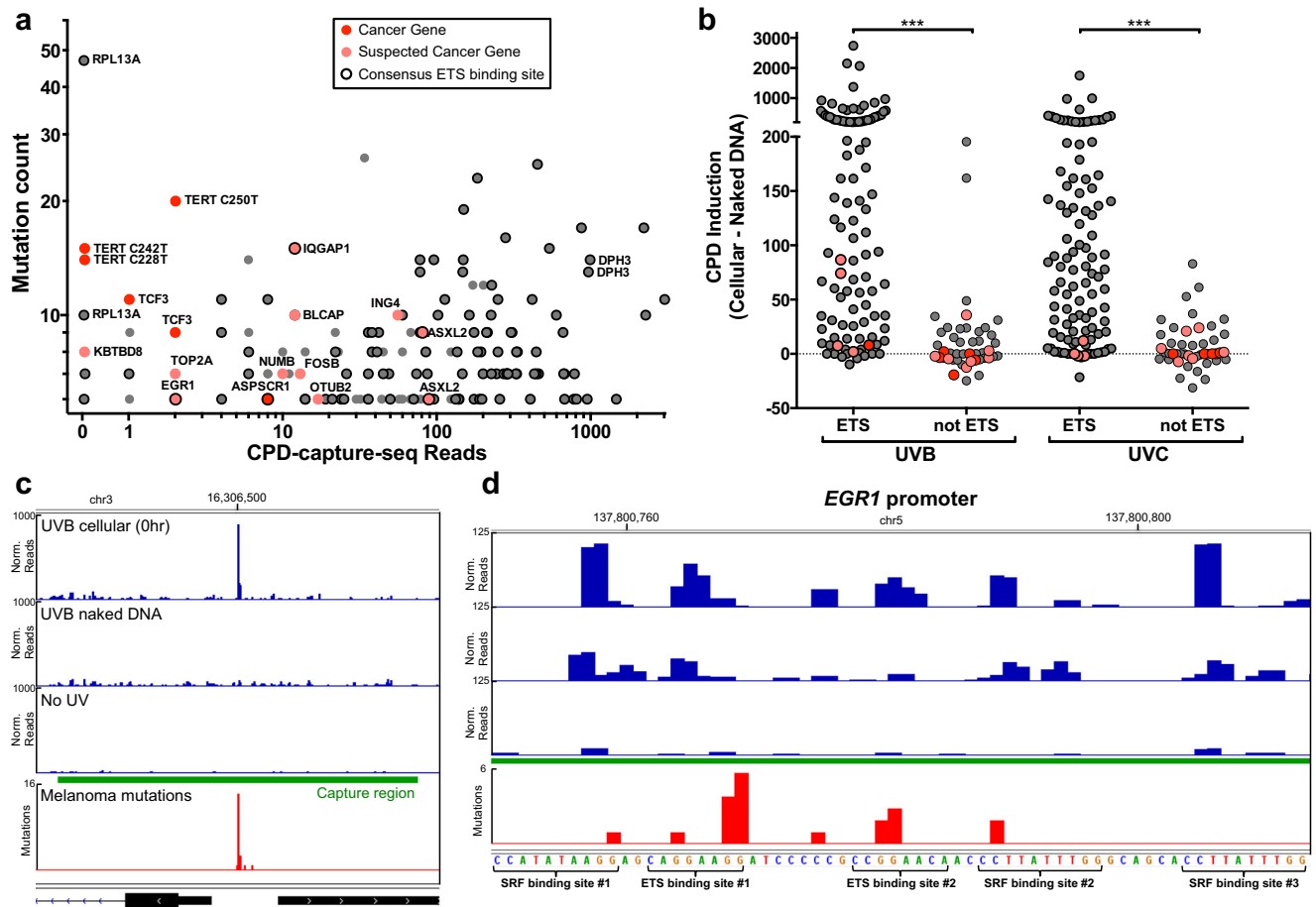

**Fig. 4 | Many recurrent promoter mutations are associated with ETS-induced UV damage hotspots. a** Plot of CPD lesions derived from CPD-capture-seq analysis of UVB-irradiated melanocytes relative to mutation count in 183 sequenced melanomas[4]. Promoter mutations associated with known or suspected cancer genes are indicated in red and salmon color, respectively. Black outline indicates that the mutation occurs in an ETS binding site. **b** Same as **a**, except the CPD induction is depicted, defined as the difference in CPD levels between UV-irradiated melanocytes and scaled UV-irradiated naked genomic DNA. Data for both UVB- and UVC-irradiated samples is depicted. ***$P < 0.001$ from two-tailed Mann-Whitney test comparing 109 (ETS) and 45 (not ETS) sites for UVB ($P < 0.0001$) and UVC ($P < 0.0001$). **c, d** Plot of normalized CPD-capture-seq reads (associated with lesion-forming dipyrimidine sites) for UVB-irradiated melanocytes or naked DNA at **c** *DPH3* (also *OXNAD1*) and **d** *EGR1* promoters. Mutation density is from 183 sequenced melanoma genomes (ICGC). Images generated using IGV[74]. Source data for graphs in **a** and **b** are provided as a Source Data file.

significantly closer to the TSS than recurrent coding mutations not associated with ETS binding motifs (median of 58,000 bp; Fig. 5e). These data indicate that recurrent coding mutations associated with ETS-induced UV damage are primarily found in the 5' end of the gene, adjacent to the promoter. This finding predicts that recurrent mutations in the 5' untranslated region (UTR), which are typically located very near the TSS, should also be associated with ETS-induced UV damage. Analysis of 79 recurrent melanoma mutations in 5'UTR regions (defined as ≥5 mutated tumors out of 183 sequenced melanomas) revealed that most of these mutations were associated with very high CPD levels in UV-irradiated melanocytes (Fig. 5f). Indeed, ~85% of recurrent 5'UTR mutations were associated with an ETS binding motif, and most showed significant UV damage induction in UV-irradiated cells relative to naked DNA (Fig. 5g). In contrast, only 9% of coding exon mutations were associated with an ETS binding motif. Taken together, these findings suggest that recurrent exon mutations associated with ETS-induced UV damage primarily occur near the TSS of genes, either in the 5'UTR or in TSS-proximal coding exon.

## Discussion

Here we have used targeted UV damage sequencing to show that many recurrent somatic mutations in melanoma are associated with, and

potentially can be explained by, localized hotspots of UV-induced CPD lesions. Notably, these include many previously identified driver mutations in melanoma, both in coding and non-coding DNA. For example, our CPD-capture-seq data indicate that seven out of twelve previously identified non-coding driver mutations in melanoma are likely recurrent passenger mutations (Supplementary Table 1), even though these mutations were identified by a sophisticated algorithm that screened for functional non-coding changes and accounted for differences in local mutation rates[4,16]. Notably, these seven promoter mutations all occurred in the promoters of housekeeping genes not previously linked to cancer (Supplementary Table 1). In contrast, the five promoter mutations that could not be explained by elevated UV damage are known non-coding driver mutations (*TERT* promoter mutations[11,12,14]) or occur upstream of a known cancer gene (*BLCAP*[41,42]) or a gene important for melanocyte differentiation (*KBTBD8*[43]).

Our CPD-capture-seq data indicates that these localized UV damage hotspots occur in UV-irradiated cells and not UV-irradiated naked DNA, and are primarily linked to ETS binding sites. Analysis of recurrent promoter mutations in melanoma revealed that ~70% of these mutation sites are associated with an ETS binding motif, suggesting that UV damage induction by ETS TFs is a major contributor to

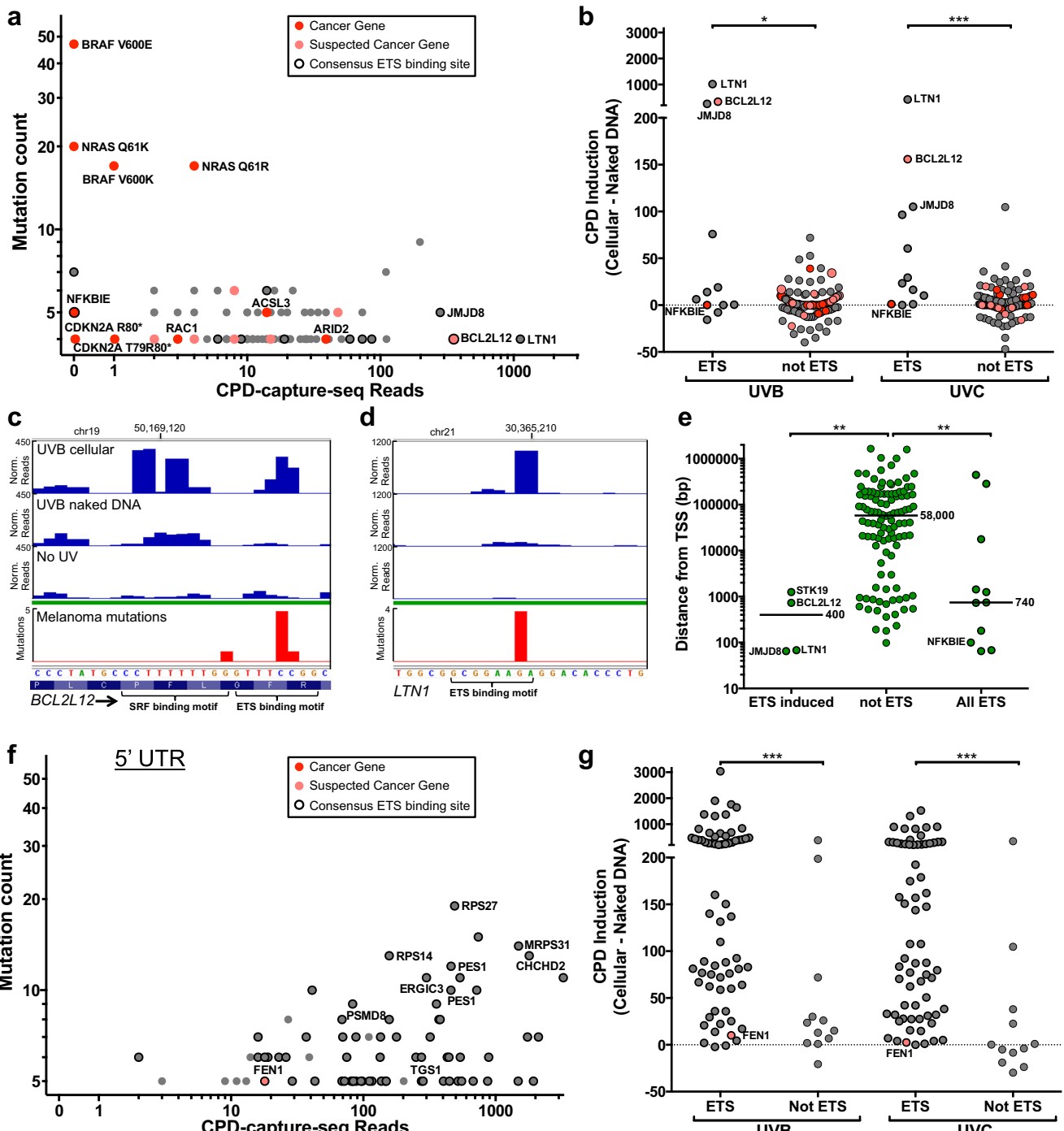

**Fig. 5 | A subset of recurrent protein-coding mutations are linked to an ETS-induced UV damage hotspot. a** Plot of mutation counts in recurrent protein-coding mutations in melanoma relative to number of CPD lesions detected by CPD-capture-seq in UVB-irradiated melanocytes. Cancer genes and suspected cancer genes are colored in red and salmon, respectively. Mutation sites overlapping with an ETS binding motif have a black outline. **b** same as **a** except CPD induction, defined as the difference in CPD-capture-seq reads between UV-irradiated melanocytes and UV-irradiated naked DNA, is shown for each mutation site for both UVB and UVC samples. ***$P < 0.001$ and *$P < 0.05$ from two-tailed Mann–Whitney test comparing 12 (ETS) and 118 (not ETS) sites for UVB ($P = 0.0418$) and UVC ($P < 0.0001$) samples. **c, d** Close-up showing normalized CPD-capture-seq data from UVB-irradiated melanocytes or naked DNA and No UV controls at the *BCL2L12* and *LTN1* genes. Mutation data from 183 sequenced melanomas from ICGC is

shown for reference. Protein-coding translation for *LTN1* is not shown because the default gene model in IGV does not include this genomic region. **e** Distance of protein-coding mutations from transcription start site (TSS) of gene. Gene coordinate data from GENCODE[79]. **$P < 0.01$ from two-tailed Mann–Whitney test comparing 110 (not ETS) sites with 4 (ETS induced; $P = 0.001$) or 11 (All ETS; $P = 0.0042$). **f** Same as **a**, except for recurrent somatic mutations in melanoma in 5' untranslated regions (5' UTR) of genes. The names of genes with a 5'UTR recurrent mutation with the greatest mutation count or that were identified as potential functional mutation in a previous publication[4] are indicated. **g** same as **b**, except for recurrent somatic mutations in melanoma in 5' untranslated regions (5' UTR) of genes. ***$P < 0.001$ from two-tailed Mann–Whitney test comparing 67 (ETS) and 12 (not ETS) sites for UVB ($P = 0.0003$) or UVC ($P < 0.0001$) data. Source data for graphs in **a, b, e–g** are provided as a Source Data file.

the mutational landscape of skin cancers. However, our data also indicate that not all ETS binding motifs are associated with UV damage induction in UV-irradiated skin cells. Indeed, many recurrently mutated ETS binding sites associated with known or suspected cancer genes (e.g., *EGR1*[51], *IQGAP1*[52], and *NFKBIE*[19]) have low CPD levels in UV-irradiated melanocytes. The lack of UV damage induction at these and other sites likely reflects the fact that they are not bound by an ETS TF in primary melanocytes. Consistent with this hypothesis, we observed differences in UV damage induction at ETS binding sites between primary melanocytes and immortalized skin fibroblasts (Supplementary Fig. 7), presumably reflecting differences in ETS TF occupancy in these cell types. This could potentially explain why a recurrently mutated ETS binding site upstream of the ribosomal protein gene *RPL13A* did not show damage induction in primary melanocytes (or fibroblasts), despite multiple lines of evidence suggesting this is a recurrent passenger mutation[9,26,31]. It is possible that an ETS TF binds this site at a later stage in melanomagenesis to promote UV damage induction and mutagenesis.

An important implication of these findings is that CPD-capture-seq can be used as a high-resolution, quantitative method for mapping TF occupancy at DNA binding sites of ETS and potentially other TFs that induce UV damage (e.g., SRF and CTCF[38,53]). This would be an especially powerful approach for mapping ETS TF binding sites, which has proven challenging using traditional methods like ChIP-seq due to the large number of ETS family TFs (28 members) that have very similar DNA binding motifs[30,54]. Indeed, our CPD-capture-seq data identified new ETS binding sites in the promoter of *PDCD11* and elsewhere in the genome.

Our data also reveal that UV damage induction at putative ETS binding sites can explain a number of recurrent protein-coding mutations. Most notable of these is *STK19* D89N, which has been identified as a recurrent driver mutation in both melanoma[1] and non-melanoma skin cancers[49], but whose functional significance is controversial[45–47,55]. Our CPD-capture-seq data indicate that CPD levels are induced at this site, consistent with biochemical data indicating that ETS1 protein can bind this region of the *STK19* gene and induce UV damage formation in vitro. Similarly, ETS-induced UV damage is associated with recurrent synonymous mutations in the *JMJD8* and *BCL2L12* genes. While this latter mutation (i.e., *BCL2L12* F17F) has been suggested to play a functional role in carcinogenesis by disrupting a potential microRNA target site[18], our CPD-capture-seq data suggest further investigation is warranted. A common feature of these coding mutations is that each occurs near the beginning of the gene, suggesting that ETS TFs primarily bind to exon sites that are located near the promoter. This is supported by the finding that many recurrent mutations in 5′UTR regions were associated with a UV damage hotspot, consistent with the observation that ~85% of these recurrent mutations occurred in an ETS motif.

Our results suggest that a similar experimental strategy could be used to identify recurrent passenger mutations in other cancer types. A recent report used the propensity of APOBEC cytidine deaminase enzymes to damage DNA at hairpin-forming sequences as a means to distinguish mutations caused by APOBEC activity from driver mutations in cancer genomes[56]. Genome-wide methods have been developed to map other types of DNA damage, including DNA alkylation[57,58], oxidative lesions[59–61], and cisplatin adducts[62], so it may be feasible to adapt this capture-sequencing strategy to investigate whether other forms of DNA damage cause recurrent mutations in different cancer types. While we have focused on measuring initial damage formation, it is also clear that DNA repair inhibition is also associated with elevated mutation rates in cancers[28,37,38,63–68]. It will be important in future studies to investigate whether targeted DNA damage sequencing can also be used to measure repair rates at potential sites of recurrent passenger mutations in a variety of different cancers.

## Methods

### Culture and UV treatment of cell lines
Normal human epidermal melanocyte (NHEM 2) cells (C-12402, PromoCell) were grown to ~80% confluence in Melanocyte culture medium (LL-0027, Lifeline cell technology) at 37 °C and 5% $CO_2$. For UV irradiation, the culture medium was removed, washed once with 1× phosphate buffered saline (PBS). The cells were then layered with 2 ml sterile PBS and irradiated with either 2500 J/m$^2$ UVB or 500 J/m$^2$ UVC light. Following irradiation, PBS was removed, cells were harvested with trypsin, collected by centrifugation and pellets were stored at −80 °C till genomic DNA isolation. Cells from plates without UV treatment were pelleted for "No UV" control and "naked DNA" control samples.

Normal human fibroblast (NHF1) cells, telomerase-immortalized[25,69] (originally derived by Dr. William Kaufmann, University of North Carolina) were grown to ~80% confluence in Dulbecco's modified Eagle's medium (DMEM) containing 10% fetal bovine serum (FBS) at 37 °C and 5% $CO_2$. The UV irradiation and cell collection procedures are similar to NHEM 2 cells, except the dose of 500 J/m$^2$ UVB or 100 J/m$^2$ UVC was used for irradiation, similar to our previous study[25]. Higher UV doses were used for melanocytes due to a previous report suggesting that this cell type may have a higher background in damage mapping experiments[27].

### Genomic DNA isolation and UV irradiation of naked DNA
Genomic DNA was isolated from the cell pellets stored at −80 °C using GenElute Mammalian genomic DNA miniprep kits (G1N70, Sigma-Aldrich). For naked DNA control, the isolated DNA was spotted on clean microscope cover glass and then exposed to UV light. A dose of 2500 J/m$^2$ UVB or 400 J/m$^2$ UVC was used for genomic DNA from melanocytes and a dose of 500 J/m$^2$ UVB or 80 J/m$^2$ UVC was used for DNA isolated from NHF1 cells. After irradiation, the DNA was collected and processed for CPD-seq library preparation.

### CPD-seq library preparation and capture sequencing
CPD-seq library preparation was carried out following published protocols[34,70] with modifications in the adapter sequences to make them suitable for Illumina sequencing. The UV-irradiated DNA was sonicated, ligated to F1 adapter, and treated with terminal transferase (M0315S, NEB). The DNA was then digested with T4 endonuclease V (T4 PDG, M0308S, NEB) and AP endonuclease (M0282S, NEB) to create 3′-OH groups immediately upstream of the CPD lesions, and the resulting fragments were ligated to a biotin-labeled second adapter, S2. The single-stranded DNA was eluted with streptavidin beads and the final PCR was done with F1 primer and different RAPID primers to barcode different samples.

F1-top 5′-GTGACTGGAGTTCAGACGTGTGCTCTTCCGATCT-3′

F1-bottom 5′-phosphate-GATCGGAAGAGCACACGTCTGAACTCCAGTCA-dideoxycytidine-3′

S2-top 5′-biotin- GACACTCTTTCCCTACACGACGCTCTTCCGATCTNNNNNN-C3-phosphoramidite-3′

S2-bottom 5′-biotin- AGATCGGAAGAGCGTCGTGTAGGGAAAGAGTGT--dideoxycytidine-3′

RAPID1 5′-CAAGCAGAAGACGGCATACGAGATTAGAACACGTGACTGGAGTTCAGACGTGTGCTCTTCCGATC-3′

RAPID2 5′-CAAGCAGAAGACGGCATACGAGATGAGCCAATGTGACTGGAGTTCAGACGTGTGCTCTTCCGATC-3′

RAPID3 5′-CAAGCAGAAGACGGCATACGAGATCAGATCTGGTGACTGGAGTTCAGACGTGTGCTCTTCCGATC-3′

RAPID4 5′-CAAGCAGAAGACGGCATACGAGATTGTGAAGAGTGACTGGAGTTCAGACGTGTGCTCTTCCGATC-3′

RAPID5 5′-CAAGCAGAAGACGGCATACGAGATACAGTGGTGTGACTGGAGTTCAGACGTGTGCTCTTCCGATC-3′

RAPID6 5′-CAAGCAGAAGACGGCATACGAGATTGAACTGGGTGACTGGAGTTCAGACGTGTGCTCTTCCGATC-3′

RAPID7 5′-CAAGCAGAAGACGGCATACGAGATTCTACGACGTGAC
TGGAGTTCAGACGTGTGCTCTTCCGATC-3′

RAPID8 5′-CAAGCAGAAGACGGCATACGAGATGCAATCCGGTGAC
TGGAGTTCAGACGTGTGCTCTTCCGATC-3′

RAPID9 5′-CAAGCAGAAGACGGCATACGAGATTCCGTCTTGTGAC
TGGAGTTCAGACGTGTGCTCTTCCGATC-3′

RAPID10 5′-CAAGCAGAAGACGGCATACGAGATTCAGGAGGGTGA
CTGGAGTTCAGACGTGTGCTCTTCCGATC-3′

RAPID0 5′-AATGATACGGCGACCACCGAGATCTACACAGGCTATA
ACACTCTTTCCCTACACGACGCTCTTCCGATCT-3′

F1 primer 5′-GTGACTGGAGTTCAGACGTGTCGTCTTCCGATCT-3′

Capture sequencing was performed by the company Rapid Genomics using the provided CPD-seq libraries (>200 ng each). A custom capture panel was designed to capture 720 bp genomic regions centered on either an active ETS binding site, defined as ETS binding site identified by ChIP-seq data for ETS family members ETS1, ELK4, or GABPA that is present in a melanocyte DNase hypersensitivity region, as previously described, or a recurrent somatic mutation in either coding sequence, promoter, 5′UTR, 3′UTR, or transcription factor binding sequence. In some cases (e.g., *STK19* or *TERT*), a larger region was captured (see Fig. 1b for more details). Typically, eleven or twelve 120mer capture probes were designed to tile across the 720 bp genomic region. Capture probes were designed to attempt to mitigate cross-hybridization with other genomic regions.

## CPD-capture-seq data analysis

CPD-capture-seq data were analyzed as previously described for CPD-seq data[34]. Briefly, the CPD-capture-seq data were aligned to the human genome (hg19) using Bowtie2[71] with default parameters. The resulting SAM file was converted to a BAM file using SAMtools[72] and then to a BED file using BEDtools[73]. Custom perl scripts were used to only retain CPD-capture-seq reads associated with putative CPD lesions at lesion-forming dipyrimidine sequences. IGV tools[74] was used to convert the resulting BED files to WIG files or TDF files, which were used for all subsequent analysis. Each CPD-capture-seq read was assigned to both positions (i.e., bases) in the associated dipyrimidine sequence, as previously described[34].

Analysis of CPD levels at active ETS or Fos/Jun binding sites (see above) was performed as previously described[25] using custom Perl scripts. UV-irradiated naked DNA data was scaled so that it had a similar number of CPDs (i.e., CPD-capture-seq reads at dipyrimidine sequences) as the matched UV-irradiated cellular data.

Cluster analysis was performed using custom Perl scripts and Treeview[75], similar to our previously described analysis[76], except data was typically analyzed at single nucleotide resolution. UV-irradiated naked DNA data was scaled so that it had the same overall number of reads at lesion-forming dipyrimidine sequences as the matched UV-irradiated cellular data. For cluster analysis, we excluded ETS binding sites that had low capture efficiency, defined as fewer than 1 lesion site per base pair in DNA flanking the ETS binding site. Z-score analysis of CPD induction was performed by calculating the average and standard deviation of CPD induction for regions flanking (6 to 180 bp from binding site midpoint) variant ETS binding sites (i.e., −3/−4 dipyrimidine). All CPD-forming positions were included in this analysis. The CPD induction at each position was transformed by subtracting the average flanking CPD induction and dividing by the standard deviation to compute the Z-score.

Poisson regression analysis was performed by comparing the sum of mutations in melanoma at positions −3/−4 of variant ETS binding sites (dependent variable) to the count of CPD-capture-seq at the same positions in UVB-irradiated naked DNA and/or UVB-irradiated melanocytes (independent variables). ETS binding sites that had low capture efficiency, defined as fewer than 1 lesion site per base pair in DNA flanking the ETS binding site, were excluded. The null model only included CPD counts for UV-irradiated naked DNA, while the alternative model included CPD counts from both UV-irradiated naked DNA and UV-irradiated melanocytes. The likelihood ratio test was used to determine if inclusion of cellular (i.e., melanocyte) CPD counts significantly improved the model. Analysis was done using GraphPad Prism (version 8). The alternative Poisson regression model had a coefficient of 0.001182 (95% CI: 0.001034 to 0.001331) for cellular CPD counts and a coefficient of −0.01532 (95% CI: −0.01790 to −0.01283) for naked DNA CPD counts.

Analysis of CPD-capture-seq read counts at recurrent mutation sites was performed using custom Perl scripts, and again analyzing only CPD-capture-seq reads associated with lesion-forming dipyrimidine sequences. Differences in CPD-capture-seq reads between matched cellular and naked DNA samples were performed after scaling the naked DNA so that it had a similar number of CPDs (i.e., CPD-capture-seq reads at dipyrimidine sequences) as the matched UV-irradiated cellular data, prior to computing the difference between the data sets. Significant differences in CPD induction between the cellular and matched naked DNA samples were determined using a Mann-Whitney test in GraphPad Prism. IGV[74] was used to visualize TDF files of different CPD-capture-seq data sets, which were normalized so that each data set depicted had an equivalent sequencing depth. Again, only CPD-capture-seq reads associated with lesion-forming dipyrimidine sequences are shown.

## Analysis of DNase-seq data

We obtained WIG files containing data for DNase-seq data for skin melanocytes (E059) from the Gene Expression Omnibus (GEO accessions GSM774243, GSM774244, and GSM1024610)[36]. We averaged the count of DNase-seq reads for each data set within 50 bp of the midpoint of an ETS binding site, and summed the averages between data sets. Data were plotted using Treeview[75]. Spearman correlation analysis was performed using GraphPad Prism on 324 ETS binding sites.

## Analysis of somatic mutations from melanoma genomes

Genome-wide maps of somatic mutation density in 183 melanoma genomes were obtained from the International Cancer Genome Consortium (ICGC) website (https://dcc.icgc.org/releases/release_20/Projects/MELA-AU) and were analyzed as previously described[25] to generate WIG and TDF files for subsequent analysis. Only data for single nucleotide variants was analyzed using IGV and at ETS binding sites. Lists of recurrent mutations at promoters, protein-coding regions, and 5′ and 3′UTRs were obtained from the published study describing these data[4]. These mutation counts were used for the analysis of recurrent driver, promoter, coding, and 5′UTR mutations. Note that this recurrent mutation counts only included identical mutations (e.g., all C > T) at the mutation sites. We used a custom Perl script to annotate whether the recurrent mutation site overlapped with an ETS binding motif (i.e., TTCCG or CTTCC), either in the −4, −3, 0, or +1 position relative to the ETS binding motif midpoint.

## Analysis of ETS1-induced CPD formation at *STK19* D89 coding sequence in vitro

The recombinant DNA binding domain of transcription factor ETS1 protein was purified, as previously described[25]. Briefly, BL21*(DE3) *E. coli* harboring murine Ets-1ΔN280 was induced at $OD_{600} = 0.6$ with 0.5 mM IPTG at 30 °C for ~4 hr. The harvested pellet was lysed by sonication and partially purified on Co-NTA resin, followed by thrombin cleavage to remove the C-terminal His×6 tag. The protein was polished on Sepharose SP (Cytiva) and eluted on a NaCl gradient. Purified protein was homogeneous as judged by Coomassie-stained SDS-PAGE. Protein concentration was determined by UV absorption at 280 nm based on the extinction coefficient 39,880 $M^{-1}$ $cm^{-1}$.

Oligonucleotides containing STK19 D89 sequence with ETS-binding motif were synthesized and PAGE-purified by IDT (Integrated DNA Technologies). CPD formation at an ETS1 binding site was analyzed as described previously[25]. Briefly, the oligonucleotide STK19-RVS (5′-CCTGAAAATAGGGTCTTCCGGCGCAGAGCA-3′) was 5′end labeled with [γ$^{32}$P]-ATP (Perkin Elmer) using T4 polynucleotide kinase (M0201S, NEB). The labeled oligonucleotide was purified using Illumina spin columns and 100 picomoles was used for annealing with equal amounts of STK19-FWD (5′-Biotin-TGCTCTGCGCCGGAAGA CCCTATTTTCAGG −3′). The annealed oligo was bound with ETS1 protein, and the binding was determined by electrophoretic mobility shift assays. The unbound and bound oligos were exposed to 1800 J/m$^2$ of UVC. The DNA was extracted with phenol:chloroform:isoamyl alcohol and pelleted using 100% ethanol. The DNA was then washed with 70% ethanol and dissolved with water and digested with T4 PDG (M0308S, NEB). The reaction was stopped with addition of formamide to the samples which were heated at 95 °C for 10 min. The samples were loaded on to a prerun 15% denaturing urea sequencing gel. Electrophoresis was carried out at 60 watts for 2 h and 10 min. The gel was exposed to a phosphor screen for 2 h and then scanned using Typhoon phosphoimager (GE Healthcare). The intensity of the bands was quantified using ImageQuant software and the sizes of the fragments were determined using radiolabeled marker oligonucleotides.

Marker oligonucleotides:
13 bases
5′-CCTGAAAATAGGG-3′
14 bases
5′-CCTGAAAATAGGGT-3′
15 bases
5′-CCTGAAAATAGGGTC-3′
16 bases
5′-CCTGAAAATAGGGTCT-3′
17 bases
5′-CCTGAAAATAGGGTCTT-3′
18 bases
5′-CCTGAAAATAGGGTCTTC-3′

**Reporting summary**

Further information on research design is available in the Nature Portfolio Reporting Summary linked to this article.

## Data availability

The CPD-capture-seq data described in the study have been deposited in the Gene Expression Omnibus (GEO) under the accession GSE225362. Previously published DNase-seq data for skin melanocytes (E059) is available from GEO under the accessions GSM774243, GSM774244, and GSM1024610. Genome-wide somatic mutation data for 183 melanoma genomes are available from the International Cancer Genome Consortium (ICGC) (https://dcc.icgc.org/releases/release_20/Projects/MELA-AU). Lists of recurrent mutations at promoters, protein-coding regions, and 5′ and 3′UTRs are available at https://www.nature.com/articles/nature22071. Source data are provided in this paper.

## Code availability

Software code is freely available at: https://github.com/bmorledge-hampton19/CPD-Capture-seq[77].

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

## Acknowledgements

We thank Dr. Steven Roberts for their helpful suggestions and assistance. We are grateful to Scott Stevison and Benjamin Morledge-Hampton for their bioinformatics assistance. We are grateful to the International Cancer Genome Consortium (ICGC) for making mutation calls from sequenced cancer genomes publically available. This research was supported by National Institute of Environmental Sciences (NIEHS) grants R01ES028698 (J.J.W.), R01ES032814 (J.J.W.), R21ES029655 (J.J.W. and G.M.K.P.), and R21ES035139 (J.J.W. and G.M.K.P.), by National Heart, Lung, and Blood Institute grant HL155178 (G.M.K.P.) and by National Science Foundation grant MCB 2028902 (G.M.K.P.).

## Author contributions

K.S., S.S., G.M.K.P., and J.J.W. designed the research and interpret results; K.S. performed CPD-capture-seq experiments; K.S. and S.S. performed in vitro ETS1 experiments; G.M.K.P. purified the ETS1 protein; K.S., S.S., G.M.K.P., and J.J.W. wrote and edited the paper.

## Competing interests

The authors declare no competing interests.
