## [Peer Review File · Nature Communications]

Detecting recurrent passenger mutations in melanoma by targeted UV damage sequencingREVIEWER COMMENTS

Reviewer #1 (Remarks to the Author):

The article Detecting recurrent passenger mutations in melanoma by targeted UV damage sequencing by Selvam et al describes a new experimental approach to detect genomic sites that accumulate recurrent CPDs through capture and sequencing at high depth. Exploring a set of (mostly non-coding, but also exonic) genomic regions they identify CPD formation hotspots, most of which they confirm (or suspect) to be related with the binding of transcription factors. Contrasting these CPD hotspots with melanoma mutational hotspots, they are able to qualitatively distinguish sites with high mutational burden due to positive selection in tumorigenesis from others resulting from accumulation of CPDs. The article is clearly and nicely written, timely and makes for a very interesting reading.

1. Is there a difference (as it appears to emerge from Fig. 1c) in sequence specificity of CPDs between cellular and naked DNA (i.e., TT/CT dipyrimidines) or is only due to experimental variability? Maybe the authors could explore the trinucleotide specificity profile of CPDs in both environments and compare both vectors component to component.
2. To further support the claim that ETS binding sites with no CPD induction at positions -4/-3, -1/0 and 0/1 (Fig. 2C, Supp. Fig 2) correspond to inactive sites, they could examine their DHS profiles in melanocytes and fibroblasts.
3. While throughout their article the authors focus on sites with positive CPD induction, they find plenty evidence of sites with negative values (Fig. 2C, for instance). What could be the mechanism behind these sites with less CPDs in cellular than naked DNA: are they protected by protein (Tfs, nucleosomes or other) binding?
4. It would be nice if the authors could provide some sense of statistical significance of the induction values observed at specific sites (i.e., a measure of how unexpected, given the distribution of CPD-capture-seq reads across cellular and naked DNA, is the positive or negative induction value observed at the site). This could aid in the subsequent analysis and discussion of passenger vs driver mutational hotspots.
5. In Figures 3a and 5a, it would be helpful if in the case of cancer genes with several mutational hotspots (e.g., BRAF, NRAS) the authors distinguish between these hotspots. For example, NRAS12, NRAS61, etc.
6. The authors point out (page 12) that low CPD induction at mutational hotspots is an evidence of positive selection. While this may indeed be true for some cases, positive selection is not the only explanation for these mutational hotspots. Low NER activity at areas overlapping these specific sites may also account for high mutational burden.

I find the method presented by the authors and in particular the results obtained with its application to study particular potential driver mutations in melanomagenesis. I wonder, however, about the feasibility to extend it to cover larger genomic regions, while including other types of relevant mutagenic UV photoproducts. Furthermore, while the authors consider in the discussion that the study could be extended to other types of damage it is worthwhile noting that their detection still lags behind that of UV photoproducts. Overall, I think the manuscript will be very interesting for a wide range of scientists working in the crossroads between genomics, DNA damage and cancer biology.

Reviewer #2 (Remarks to the Author):

The manuscript by Selvam et. al., described the new technology - CPD-capture-seq. Authors used this method to measure susceptibility to UV lesions in EST binding sites and in recurrently mutated positions in melanoma.

Overall, CPD density for more than 3 MB of the genome has been characterized. High sequencing depth and single nucleotide resolution deliver CPD hotspots. The discrepancy between CPD maps and melanoma mutation density was used to separate passenger hotspot positions and true driver mutations. The paper is generally well written and data of good quality.

I think the paper presented primarily as a new method, secondary as a resource, and third as an analysis. Still, a huge portion of the manuscript discusses biological results.

My comments are solely about quantitative analysis.

1. It is unintuitive to use the subtraction of CPD rate in vivo and CPD rate in protein-free DNA as a metric of UV susceptibility. CPD density in cellular DNA should be used as a predictor because this is the measure of DNA damage.

2. One of the most important characterizations of CPD approach should be explained mutation rate variance in melanoma. While the data suggests that the track has some predictive power, the relation between CPD density and melanoma mutation rate was not quantified. One of the ways to estimate the fraction of variance that could be explained by CPD density would be to calculate pseudo R2 for Poisson regression on the number of melanoma mutations and use the log-likelihood of CPD density for naked DNA as a null.

DNA positions that were selected as recurrent mutations have to be filtered during regression training because these sites are subjected to ascertainment bias. It is also possible to include trinucleotide context in the regression approach as an interaction term. CPD lesions in different contexts could be more or less mutagenic, e.g. TT dimers are rarely causing mutations. The inclusion of trinucleotide contexts could address this question.

3. With the regression approach, it would be possible to compare expected vs observed number of mutations for any category of sites: e.g. coding nucleotides surrounding recurrent mutations, recurrent coding mutations itself, recurrent non-coding mutations, EST sites, and surrounding sites.

With the expected number of mutations available, it would be possible to swap CPD-density with the expected number of mutations in Figures 4,5 and 6.

4. Thresholds for recurrence are different throughout the paper. The choice of the threshold has to be explained and formally justified. Examples of inconsistent thresholds:

we performed similar analysis on all recurrent protein-coding mutations in the melanoma cohort (defined as ≥ 4 mutated tumors out of 183 sequenced melanomas)

Analysis of 79 recurrent melanoma mutations in 5'UTR regions (defined as ≥ 5 mutated tumors out of 183 sequenced melanomas)

And Fig. 1b

5. The authors found that CPD density varies among active EST binding sites. It could be interesting to test the correlation between the strength of DHS footprint and the density of CPD lesions.

6. This study implies that susceptibility to UV lesions is a major determinant of mutation rate in melanoma at single nucleotide resolution. It would be great to discuss the role of DNA repair and provide a literature overview on the topic.

Minor comment:

In Fig. 1d density plot/histogram for each category of sites, like EST, TFBS, exons ... will make the figure more accessible.

Response to Reviewers

Reviewer #1:

1. “Is there a difference (as it appears to emerge from Fig. 1c) in sequence specificity of CPDs between cellular and naked DNA (i.e., TT/CT dipyrimidines) or is only due to experimental variability? Maybe the authors could explore the trinucleotide specificity profile of CPDs in both environments and compare both vectors component to component.”

RESPONSE: We analyzed the dinucleotide frequency of the CPD-capture-seq data in UV irradiated cells relative to naked DNA in UVC-irradiated melanocytes and UVB- and UVC-irradiated normal human fibroblasts. The results (see Supplementary Fig. 2) indicate that there is some experimental variability in the relative dinucleotide frequency between the cellular and naked DNA samples in the different experiments. However, the ratio of TC/CT reads seem to be different between the cellular and naked DNA samples, as pointed out by the Reviewer.

To investigate this further, we analyzed the frequency of CPD-capture-seq reads associated with different flanking DNA sequence contexts (e.g., CTTCCG, underline indicates the location of the CPD lesion) in the cellular and naked DNA samples, as suggested by the Reviewer. This analysis identified a number of sequence contexts in which the number of CPD-capture-seq reads was elevated in the UV-irradiated cells relative to the UV-irradiated naked DNA control (Supplementary Fig. 9). These include a number of sequence contexts containing TC, CC, and TT dipyrimidines. Notably, closer inspection of some of these sequence contexts indicate that they likely are associated with ETS binding sites (e.g., CTTCCG, TTCCGG, and TTCCGC), consistent with our findings that ETS binding promotes CPD formation at TC and CC dinucleotides in its binding site. A number of sequence contexts also had a smaller number of CPD-capture-seq reads in the UV-irradiated cells relative to the naked DNA control. Many of these were sequence contexts containing CT dipyrimidines, including some that also matched the ETS binding consensus (e.g., CACITC, TACITC, CGCTTC, and CCCTTC), again consistent with our findings that ETS binding suppresses CPD formation at a CT dinucleotide in its binding site (see Fig. 2).

Interestingly, we also observed CPD enrichment in cells at sequence contexts that matched the binding consensus (CCAAT/ATTGG) of the NF-Y transcription factor (e.g., GATTGG, CATTGG, TATTGG, and AATTGG), consistent with a previous report that NF-Y binding induces CPD formation at a TT sequence in its binding motif (Hu *et al.* 2017). In contrast, we observed CPD depletion at GACTCA sequences, which match the binding motif of Fos/Jun transcription factors and have been previously reported to suppress CPD formation (MAO *et al.* 2018). Since the majority of capture regions targeted promoters and other regulatory regions that contains ETS binding sites, it is possible that CPD modulation by ETS, NF-Y, Fos/Jun and potentially other transcription factors can partly explain the observed changes in dinucleotide frequencies in the CPD-capture-seq data. These new data are shown in Supplementary Fig. 9 and mentioned on page 10 of the revised manuscript.

2. “To further support the claim that ETS binding sites with no CPD induction at positions -4/-3, -1/0 and 0/1 (Fig. 2C, Supp. Fig 2) correspond to inactive sites, they could examine their DHS profiles in melanocytes and fibroblasts.”

RESPONSE: We have previously shown that in UV-irradiated cells, CPD induction is observed at active ETS binding sites associated with accessible DNase I hypersensitivity sites (DHS), but little to no CPD induction is present at inactive ETS binding sites not associated with DHS (MAO *et al.* 2018). The ETS binding sites selected for capture were all associated with a DHS. Hence, the melanocyte CPD-capture-seq data shown in Fig. 2C suggests that not all ETS binding sites located in a DHS are necessarily bound by an ETS transcription factor.

We analyzed melanocyte DNase-seq data associated with each active ETS binding site containing a -4/-3 dipyrimidine (e.g., Fig. 2C), as suggested by the Reviewer. This analysis indicated that while there were abundant DNase-seq reads at all binding sites, consistent with these binding sites being associated with the DNase hypersensitivity site, the count of DNase-seq reads did significantly vary between binding sites (Supplementary Fig. 7). Importantly, the average DNase-seq read count significantly correlated with CPD induction at ETS binding sites (Supplementary Fig. 7; $P < 0.0001$), as sites with no or little CPD induction were associated with fewer DNase-seq reads. Taken together, these findings confirm that even among putatively active sites, lower CPD induction correlates with lower DNase I accessibility. These new findings are discussed on page 9, paragraph 2 of the revised manuscript.

3. “While throughout their article the authors focus on sites with positive CPD induction, they find plenty evidence of sites with negative values (Fig. 2C, for instance). What could be the mechanism behind these sites with less CPDs in cellular than naked DNA: are they protected by protein (Tfs, nucleosomes or other) binding?”

RESPONSE: We primarily focused on positive CPD induction since that is more straightforward to detect using our CPD-capture-seq data. The Reviewer is correct that nucleosomes and certain transcription factors suppress CPD formation. We have previously shown that the molecular distance and relative torsion angle between the lesion-forming C5-C6 double bonds of neighboring pyrimidines play an important role in determining the frequency of CPD formation. For example, we have recently shown that DNA binding by the ETS transcription factor increases the distance and torsion angle to unfavorable positions at the C₋₃T₋₂ position in the ETS binding motif, which translates to lower CPD formation at these positions in UV-irradiated cells (MAO *et al.* 2018). This mechanism can explain some of the negative values observed in the middle of the ETS binding sites in Figure 2C. Similarly, the bending of DNA at minor-in rotational settings in nucleosomes results in unfavorable distance and torsion angles between neighboring pyrimidine bases (STARK *et al.* 2022), again causing lower CPD formation.

Analysis of CPD induction in different hexamer sequences indicated that CPD formation is suppressed at GACTCA sites (see response to point 1 above). Since this matches the sequence motif bound by Fos/Jun transcription factors, we analyzed CPD enrichment in UVB-irradiated melanocytes relative to naked DNA control at active Fos/Jun (i.e., AP-1) binding sites that overlapped with our CPD-capture-seq regions

(Supplementary Fig. 9). This analysis indicated that CPD formation is suppressed at Fos/Jun binding sites, consistent with our previous report. These new data are discussed on pages 10-11 of the revised manuscript.

4. “It would be nice if the authors could provide some sense of statistical significance of the induction values observed at specific sites (i.e., a measure of how unexpected, given the distribution of CPD-capture-seq reads across cellular and naked DNA, is the positive or negative induction value observed at the site). This could aid in the subsequent analysis and discussion of passenger vs driver mutational hotspots.”

RESPONSE: We calculated the average and standard deviation of CPD induction (i.e., difference in UV-irradiated cellular versus naked DNA CPD-capture seq reads) for regions flanking variant ETS binding sites (i.e., 6 to 180 base pairs away from ETS binding site midpoint). Similar values were obtained if we analyze CPD induction for flanking DNA of all CPD-capture-seq regions. We used these values, which are mentioned on pages 7-8 of the revised manuscript, to calculate the Z-score of CPD induction at ETS binding sites. This analysis indicates that CPDs are highly induced (Z-score > 3) at many ETS binding sites. These new data are shown in Supplementary figure 6 and described on pages 7-8 of the revised manuscript.

5. “In Figures 3a and 5a, it would be helpful if in the case of cancer genes with several mutational hotspots (e.g., BRAF, NRAS) the authors distinguish between these hotspots. For example, NRAS12, NRAS61, etc.”

RESPONSE: We thank the Reviewer for this suggestion. For driver genes with multiple hotspot mutations (e.g., BRAF V600E and V600K), we now indicate the particular amino acid change to distinguish between each hotspot (see revised Fig. 3a, 4a, and 5a).

6. “The authors point out (page 12) that low CPD induction at mutational hotspots is an evidence of positive selection. While this may indeed be true for some cases, positive selection is not the only explanation for these mutational hotspots. Low NER activity at areas overlapping these specific sites may also account for high mutational burden.”

RESPONSE: We agree that low NER activity could also potentially explain some of the observed mutational hotspots. In the revised manuscript, we now discuss this important point on page 20, paragraph 2.

7. “I find the method presented by the authors and in particular the results obtained with its application to study particular potential driver mutations in melanomagenesis. I wonder, however, about the feasibility to extend it to cover larger genomic regions, while including other types of relevant mutagenic UV photoproducts. Furthermore, while the authors consider in the discussion that the study could be Supplementary to other types of damage it is worthwhile noting that their detection still lags behind that of UV photoproducts. Overall, I think the manuscript will be very interesting for a wide range of scientists working in the crossroads between genomics, DNA damage and cancer

biology.”

RESPONSE: We thank the Reviewer for the helpful comments. We have developed a new method known as UVDE-seq that uses CPD photolyase and ultraviolet damage endonuclease (UVDE) enzyme to specifically cleave UV-induced 6-4PPs and other atypical photoproducts (BOHM *et al.* 2022). In a future study, we hope to use the same capture-sequencing approach to map these other UV photoproducts in the same capture regions. We agree that it may be potentially difficult to extend this method to capture and sequence larger regions of the genome and map other types of DNA damage.

Reviewer #2:

“The manuscript by Selvam *et al.*, described the new technology - CPD-capture-seq. Authors used this method to measure susceptibility to UV lesions in EST binding sites and in recurrently mutated positions in melanoma. Overall, CPD density for more than 3 MB of the genome has been characterized. High sequencing depth and single nucleotide resolution deliver CPD hotspots. The discrepancy between CPD maps and melanoma mutation density was used to separate passenger hotspot positions and true driver mutations. The paper is generally well written and data of good quality. I think the paper presented primarily as a new method, secondary as a resource, and third as an analysis. Still, a huge portion of the manuscript discusses biological results. My comments are solely about quantitative analysis.”

1. “It is unintuitive to use the subtraction of CPD rate *in vivo* and CPD rate in protein-free DNA as a metric of UV susceptibility. CPD density in cellular DNA should be used as a predictor because this is the measure of DNA damage.”

RESPONSE: For our analysis, we use both CPD density *in vivo* (i.e., in UV-irradiated cells) and the difference in CPD density between UV-irradiated cells and UV-irradiated naked DNA to analyze recurrent mutation sites in melanoma. While we agree with the Reviewer that CPD density *in vivo* is an important measure of cellular damage levels, which is why we use it in our study, we believe that the difference in CPD levels between UV-irradiated cells and naked DNA is also informative for the following reasons. First, the frequency (or count) of CPD-capture-seq reads at a particular site can be potentially influenced by experimental variables in the CPD-capture-seq protocol, including the efficiency of PCR amplification, sequencing, and/or the capture efficiency of library fragments. The same experimental variables should affect the naked DNA CPD-capture-seq libraries, so calculating the difference in cellular and naked DNA reads should help mitigate their potential impact.

Second, we hypothesize that many of the CPD hotspots in cells are caused by transcription factors, such as ETS proteins, binding to DNA in cells. CPD induction by TF binding results in significantly higher CPD-capture-seq reads in UV-irradiated cells relative to the naked DNA control, which can be quantified by the difference in CPD counts. Hence, the difference in CPD levels provides important insight into the

molecular mechanism responsible for the CPD hotspot. For example, while many CPD hotspots are elevated in UV-irradiated cells relative to the naked DNA control, our analysis also indicates that some recurrent mutations are associated with similarly high damage levels in both cellular and naked DNA experiments. In these cases, it is possible that the DNA sequence context, as opposed to protein binding, may be driving elevated damage levels, which we hope to explore in a future study.

2. “One of the most important characterizations of CPD approach should be explained mutation rate variance in melanoma. While the data suggests that the track has some predictive power, the relation between CPD density and melanoma mutation rate was not quantified. One of the ways to estimate the fraction of variance that could be explained by CPD density would be to calculate pseudo R2 for Poisson regression on the number of melanoma mutations and use the log-likelihood of CPD density for naked DNA as a null. DNA positions that were selected as recurrent mutations have to be filtered during regression training because these sites are subjected to ascertainment bias. It is also possible to include trinucleotide context in the regression approach as an interaction term. CPD lesions in different contexts could be more or less mutagenic, e.g. TT dimers are rarely causing mutations. The inclusion of trinucleotide contexts could address this question.”

RESPONSE: We used Poisson regression, as suggested by the Reviewer, to analyze the relationship between CPD density in UVB-irradiated melanocytes and mutation density in melanoma. We focused on the -4/-3 position of ETS binding sites (see Fig. 2C), since these positions are frequently mutated in melanoma (thereby potentially avoiding issues with zero-inflation, etc.) and because most binding sites had a cytosine-containing dipyrimidine at this position. Hence, we analyzed CPD density (number of CPD-capture-seq reads) in UVB-irradiated melanocytes and UVB-irradiated naked DNA relative to mutation counts in melanoma at positions -4/-3 (summed) for 324 independent ETS binding sites (Fig. 2C). We compared a Poisson regression model that included both cellular and naked DNA CPD counts as independent variables (model 1, alternative hypothesis) to a simpler model that just had the naked DNA CPD counts (model 2, null hypothesis) as an independent variable. In both cases, the mutation count was the dependent variable.

The pseudo R2 for model 1 was 0.14, and there were significant (i.e., non-zero) coefficients for both cellular and naked DNA CPD counts ($P < 0.0001$). Notably, while the coefficient for cellular CPD count variable was positive, the coefficient was negative for the naked CPD count. Hence, the model was essentially using the scaled difference in cellular and naked DNA CPD counts to predict mutation counts. In contrast, the pseudo R2 for model 2 (null) was < 0.001 , and the coefficient for naked DNA CPD count was not significantly different than zero ($P > 0.05$). A likelihood ratio test rejected the null hypothesis (model 2; $P < 0.0001$), in favor of the alternative hypothesis (model 1). Taken together, these data indicate that while CPD counts of UV-irradiated naked DNA does not appear to predict mutation density at ETS binding sites, inclusion of cellular CPD counts does predict mutation counts.

Since these positions were selected simply because they contained active ETS binding sites, there should not be an issue with ascertainment bias. In future studies, we

hope to expand this model and analysis by including more complicated terms associated with trinucleotide context, etc., as suggested by the Reviewer, and to look at other genomic regions besides ETS binding sites. This analysis is discussed on page 8, paragraph 2 of the revised manuscript.

3. "With the regression approach, it would be possible to compare expected vs observed number of mutations for any category of sites: e.g. coding nucleotides surrounding recurrent mutations, recurrent coding mutations itself, recurrent non-coding mutations, EST sites, and surrounding sites. With the expected number of mutations available, it would be possible to swap CPD-density with the expected number of mutations in Figures 4,5 and 6."

RESPONSE: We agree that it would be interesting to try to use the Poisson regression model (or something similar) to try to predict mutation density and compare it with actual mutation density, and is something that we hope to do in future studies. However, we believe this sort of analysis is beyond the scope of the current manuscript.

4. "Thresholds for recurrence are different throughout the paper. The choice of the threshold has to be explained and formally justified. Examples of inconsistent thresholds: we performed similar analysis on all recurrent protein-coding mutations in the melanoma cohort (defined as ≥ 4 mutated tumors out of 183 sequenced melanomas) Analysis of 79 recurrent melanoma mutations in 5'UTR regions (defined as ≥ 5 mutated tumors out of 183 sequenced melanomas) And Fig. 1b."

RESPONSE: The thresholds were largely based on a previous study characterizing recurrent mutations in sequenced melanoma genomes (HAYWARD *et al.* 2017). In this study, the threshold of ≥ 5 mutations was used for all non-coding recurrent mutations (e.g., promoter or 5'UTR). Since we were using these mutations for our analysis, we were limited in the threshold that could be used. For the recurrent coding mutations, a much lower threshold was used to catalog mutations, which is why we used a threshold of ≥ 4 mutations for coding exon mutations. The large number of recurrent promoter mutations required that we only show sites with ≥ 6 mutations in order to make the figure legible. However, we now show an expanded version of the figure in Supplementary Fig. 12, which shows all recurrent promoter mutations (≥ 5 mutations).

5. "The authors found that CPD density varies among active EST binding sites. It could be interesting to test the correlation between the strength of DHS footprint and the density of CPD lesions."

RESPONSE: As discussed in the response to Reviewer 1 (see point 2), we analyzed melanocyte DNase-seq data for the ETS binding sites shown in Fig. 2C. This analysis indicated that local binding site accessibility, which we calculated as the number of DNase-seq reads within 50bp of the ETS binding site, showed a significant positive correlation with CPD induction at these sites ($P < 0.0001$). These findings indicate that CPD-induction associated with ETS binding is associated with binding sites in more accessible chromatin regions, as measured by DNase-seq. These new data are shown

in new Supplementary Fig. 7, and discussed on page 9, paragraph 2 of the revised manuscript.

6. "This study implies that susceptibility to UV lesions is a major determinant of mutation rate in melanoma at single nucleotide resolution. It would be great to discuss the role of DNA repair and provide a literature overview on the topic."

RESPONSE: We agree, and now discuss the potentially important role of DNA repair efficiency, which also is likely to impact mutation rates in melanoma (see page 20, paragraph 2 of the revised manuscript). We hope to use the CPD-capture-seq approach to also measure repair efficiency at a single-nucleotide resolution in a future study.

7. "Minor comment: In Fig. 1d density plot/histogram for each category of sites, like EST, TFBS, exons ... will make the figure more accessible."

RESPONSE: Given the large number of capture regions, particularly for ETS binding sites, it would be difficult to show a legible density plot/histogram. For this reason, we have not made this change.

Finally, we would like to thank each of the Reviewers for their helpful comments and suggestions. Their efforts have helped to significantly improve the manuscript.

Sincerely,

John Wyrick
(on behalf of the co-authors)

References

- Bohm, K. A., S. Sivapragasam and J. J. Wyrick, 2022 Mapping atypical UV photoproducts in vitro and across the *S. cerevisiae* genome. STAR Protoc 3: 101059.
- Hayward, N. K., J. S. Wilmott, N. Waddell, P. A. Johansson, M. A. Field *et al.*, 2017 Whole-genome landscapes of major melanoma subtypes. Nature 545: 175-180.
- Hu, J., O. Adebali, S. Adar and A. Sancar, 2017 Dynamic maps of UV damage formation and repair for the human genome. Proc Natl Acad Sci U S A.
- Mao, P., A. J. Brown, S. Esaki, S. Lockwood, G. M. K. Poon *et al.*, 2018 ETS transcription factors induce a unique UV damage signature that drives recurrent mutagenesis in melanoma. Nat Commun 9: 2626.
- Stark, B., G. M. K. Poon and J. J. Wyrick, 2022 Molecular mechanism of UV damage modulation in nucleosomes. Comput Struct Biotechnol J 20: 5393-5400.

REVIEWERS' COMMENTS

Reviewer #1 (Remarks to the Author):

The authors have satisfactorily responded all my comments. I have no further issues to raise.

The manuscript by Selvam et al. is significantly improved after revision. Just a few minor issues need to be resolved before the paper is fully ready for publication.

1. I believe that the result of Figure S. 7 about the correlation between DHS and CPD is novel, exciting, and worthy of the main text.
2. It is surprising to me that authors still want to use the difference between naked and cellular DNA instead of Poisson regression predictions or any other to analyze mutational hotspots in melanoma, especially after regression was implemented already.
3. I believe there is a misunderstanding about my suggestion about Fig. 1b. I attached a visual suggestion to clarify.

RESPONSE TO REVIEWERS

Reviewer #1:

1. "The authors have satisfactorily responded all my comments. I have no further issues to raise."

RESPONSE: We thank the Reviewer again for their helpful comments.

Reviewer #2:

1. "I believe that the result of Figure S.7 about the correlation between DHS and CPD is novel, exciting, and worthy of the main text."

RESPONSE: We agree, and in the revised manuscript have moved the DNase-seq data from Fig. S7 into the main text Fig. 2C.

2. "It is surprising to me that authors still want to use the difference between naked and cellular DNA instead of Poisson regression predictions or any other to analyze mutational hotspots in melanoma, especially after regression was implemented already."

RESPONSE: In our manuscript, we analyze mutation hotspots in melanoma using both CPD density in cellular samples (e.g., primary melanocytes) and the CPD induction in cellular samples relative to the naked DNA control (i.e., difference in CPD density). We believe that both of these metrics are helpful for analyzing potential causes of mutation hotspots in melanoma, particularly hotspots associated with ETS binding sites. The cellular CPD levels indicate the amount of damage associated with a particular site, and therefore can potentially identify mutation hotspots that are associated with, and potentially caused by, UV damage hotspots in cells. The CPD induction metric indicates whether the high damage observed in cells is intrinsic to the DNA sequence (and therefore is also present in the naked DNA), or occur specifically in cells, such as CPD induction caused by DNA binding by ETS or other transcription factors. Since one of the key conclusions of our manuscript is that damage induction due to ETS transcription factor binding likely causes damage and mutation hotspots in melanoma, it is important for us to include analysis of CPD induction in our study. Finally, the Poisson regression that we performed indicates that the scaled difference of the cellular and naked DNA samples is the best predictor of mutation density, at least at ETS binding sites, again validating this approach.

We agree that it would be interesting to use Poisson regression to analyze other mutation hotspots in melanoma (i.e., beyond the ETS binding site analysis performed in this paper), but we believe this is beyond the scope of our current study and is instead something we hope to do in a future study.

3. "I believe there is a misunderstanding about my suggestion about Fig. 1b. I attached

a visual suggestion to clarify.”

RESPONSE: We thank the Reviewer for the clarification. We have now added graphs showing the CPD density across each category of capture regions in a new panel of Figure 1 (Fig. 1e).